# Evaluation of Baculoviruses as Gene Therapy Vectors for Brain Cancer

**DOI:** 10.3390/v15030608

**Published:** 2023-02-22

**Authors:** Matías Garcia Fallit, Matías L. Pidre, Antonela S. Asad, Jorge A. Peña Agudelo, Mariana B. Vera, Alejandro J. Nicola Candia, Sofia B. Sagripanti, Melanie Pérez Kuper, Leslie C. Amorós Morales, Abril Marchesini, Nazareno Gonzalez, Carla M. Caruso, Víctor Romanowski, Adriana Seilicovich, Guillermo A. Videla-Richardson, Flavia A. Zanetti, Marianela Candolfi

**Affiliations:** 1Instituto de Investigaciones Biomédicas (INBIOMED, UBA-CONICET), Facultad de Medicina, Universidad de Buenos Aires, Consejo Nacional de Investigaciones Científicas y Técnicas, Ciudad Autónoma de Buenos Aires C1121A6B, Argentina; 2Departamento de Química Biológica, Facultad de Ciencias Exactas y Naturales, Universidad de Buenos Aires, Ciudad Autónoma de Buenos Aires C1428BFA, Argentina; 3Instituto de Biotecnología y Biología Molecular (IBBM, UNLP-CONICET), Facultad de Ciencias Exactas, Universidad Nacional de La Plata, La Plata B1900, Argentina; 4Departamento de Biología Celular e Histología, Facultad de Medicina, Universidad de Buenos Aires, Buenos Aires, Ciudad Autónoma de Buenos Aires C1121A6B, Argentina; 5Fundación Para la Lucha Contra las Enfermedades Neurológicas de la Infancia (FLENI), Ciudad Autónoma de Buenos Aires C1121A6B, Argentina; 6Instituto de Ciencia y Tecnología ‘‘Dr. Cesar Milstein”, CONICET, Saladillo 2468 (C1440FFX), Ciudad Autónoma de Buenos Aires C1428, Argentina

**Keywords:** baculoviruses, glioblastoma, astrocytes, gene therapy, brain

## Abstract

We aimed to assess the potential of baculoviral vectors (BV) for brain cancer gene therapy. We compared them with adenoviral vectors (AdV), which are used in neuro-oncology, but for which there is pre-existing immunity. We constructed BVs and AdVs encoding fluorescent reporter proteins and evaluated their transduction efficiency in glioma cells and astrocytes. Naïve and glioma-bearing mice were intracranially injected with BVs to assess transduction and neuropathology. Transgene expression was also assessed in the brain of BV-preimmunized mice. While the expression of BVs was weaker than AdVs in murine and human glioma cell lines, BV-mediated transgene expression in patient-derived glioma cells was similar to AdV-mediated transduction and showed strong correlation with clathrin expression, a protein that interacts with the baculovirus glycoprotein GP64, mediating BV endocytosis. BVs efficiently transduced normal and neoplastic astrocytes in vivo, without apparent neurotoxicity. BV-mediated transgene expression was stable for at least 21 days in the brain of naïve mice, but it was significantly reduced after 7 days in mice systemically preimmunized with BVs. Our findings indicate that BVs efficiently transduce glioma cells and astrocytes without apparent neurotoxicity. Since humans do not present pre-existing immunity against BVs, these vectors may constitute a valuable tool for the delivery of therapeutic genes into the brain.

## 1. Introduction

Glioblastomas (GBM) are the most frequent and aggressive malignant primary brain tumors in adults and are characterized for being highly aggressive and diffuse [1,2,3]. The standard of care for these patients includes neurosurgery, radiation therapy, and chemotherapy. However, the survival of GBM patients is very short due to the tumor’s highly invasive nature and intrinsic resistance to standard therapy [4]. According to the latest report from CBTRUS (the Central Brain Tumor Registry of the United States), the estimated median survival of GBM is 8 months, and the 5-year survival rate is below 7% [1]. Therefore, it is urgent to develop new treatment strategies.

Gene therapy allows therapeutic transgenes to be expressed locally in the area adjacent to the main tumor mass after surgery [5]. Local administration of gene therapy vectors, mainly adenoviral (AdV) and retroviral vectors, has been extensively evaluated in GBM patients, showing good transduction efficiency and toxicological profile [6,7]. Oncolytic virotherapy has also experienced enormous development and local treatment with oncolytic AdVs has been evaluated in several clinical trials in GBM patients. However, the efficacy of this strategy remains to be determined [8].

AdV vectors are the most widely used viral vectors in neuro-oncology and are the gold standard in gene therapy for GBM, as they include many advantages: they transduce dividing or quiescent cells, their viral genome is easily modified using recombinant DNA technology, they can be produced in high titers, [10^9^–10^13^ plaque-forming unit (PFU)/mL)], and their genome remains episomal, which reduces the risk of insertional mutagenesis [9,10,11]. Although the use of certain alternative adenoviral serotypes for gene delivery, which have lower seroprevalence, are being studied [12], AdV serotype 5 is the most widely used in clinical research (NCT03596086, NCT03603405, NCT02026271, NCT01811992). However, since virtually the entire population has pre-existing anti-AdV immunity through vaccination or natural infection, their transgene expression may not be stable, as it has been shown that the immune system clears the vector in 7–14 days after its administration, even when injected in the brain [13]. In fact, an increase in the titers of circulating anti-adenoviral antibodies is observed upon intratumoral injection with therapeutic AdVs in GBM patients [14]. Thus, it is important to increase the availability of gene therapy vectors to treat brain cancer and other neurological disorders. Larger availability of gene therapy vectors may allow the development of therapeutic combinations with improved efficacy to treat these deadly tumors.

Baculoviruses are enveloped DNA viruses that infect insects in the larval stage and may emerge as an alternative vector in neuro-oncology. Baculoviral (BV) vectors have been shown to transduce different cell types from various species, such as human and murine cells, including mesenchymal cells [15,16]. These vectors do not integrate in the host genome and are non-replicative in mammalian cells. A great advantage of BVs is their great capacity to accommodate foreign DNA, as they have a circular genome of 134 kb with an insertion capacity that easily exceeds 38 kb and can accommodate multiple gene sequences required for the assembly of large protein complexes using recombinant virus construction systems such as MultiBac and HR-Bac [17,18,19,20]. Its most salient characteristic is that no pre-existing immunity against BV has been detected in humans, and thus, BV transduction would be more stable than AdV, especially in immunologically privileged tissues such as the brain. BVs are much easier to produce in large scale than the rest of the vectors, since they easily propagate in insect cells without using specialized equipment [21,22]. Since BVs have been shown to efficiently transduce tumor cells, i.e., prostate cancer cells, pituitary, and brain cells [23,24,25,26], they may represent a valuable tool for the expression of therapeutic transgenes for the treatment of GBM and other brain disorders. In fact, it has been previously shown that BVs can transduce GBM cells [21,22]. However, up to now, no clinical trials have implemented the use of these viruses as vectors of gene therapy in any pathology. Thus, our aim was to perform an exhaustive preclinical evaluation of recombinant baculoviruses to deliver therapeutic transgenes for the treatment of brain cancer. We assessed the transduction efficiency of BVs encoding fluorescent proteins in neoplastic and normal astrocytes in vitro and in vivo and compared them with AdVs. We also characterized BV-mediated transgene expression in the brain of naïve and preimmunized mice. Our findings indicate that BVs have a significant potential to become a useful tool for transgene delivery in neuro-oncology.

## 2. Materials and Methods

### 2.1. Baculoviral Vectors

We constructed BV encoding dTomato or citrine under the control of the Cytomegalovirus (CMV) promoter. The expression cassette was cloned into an EcoRV-NotI digested transfer vector pBacPAK9 (Clontech, Mountainview, CA, USA). The AcMNPV flanking sequences allow homologous recombination with viral DNA in insect cells to transfer the expression cassette to the polyhedrin locus of viral DNA. For this, recombinant pBacPAK was co-transfected into the Trichoplusia ni insect cell line BTI-TN-5B1-4 (High Five^TM^ cells; Thermo Fisher Scientific, Waltham, MA, USA) with bApGOZA DNA. Cells were maintained in Grace’s medium (Thermo Fisher Scientific) supplemented with 10% FBS at 27 °C until signs of infection appeared. The expression of dTomato and citrine was verified by epifluorescent microscopy. In experiments where BVs were injected in vivo and compared with AdVs, vectors were purified by sucrose cushion. Briefly, vectors were clarified and transferred to an ultracentrifuge tube with a cushion of 25% sucrose in PBS. Subsequently, ultracentrifugation was performed at 80,000× *g* for 4 h at 4 °C. Finally, the supernatant was discarded, and the pellet was resuspended in PBS. For the in vitro assays, and in vivo experiments in which BVs were used alone, vectors were clarified by centrifugation at 22,000× *g* for 2 h at 4 °C. The titration was performed on a monolayer of High Five^TM^ insect cells as PFU, with these titers coinciding with the foci of infection of the dTomato and citrine reporter genes.

### 2.2. Adenoviral Vectors

We constructed a non-replicative human adenovirus serotype 5 (AdV) encoding the red fluorescent dTomato reporter gen, under the control of the CMV immediate-early promoter, using the ViraPower™ Adenoviral Expression System (Invitrogen, Thermo Fisher Scientific, Waltham, MA, USA) and following the manufacturer’s recommendations. Briefly, the 1030 bp heterologous nucleotide sequence, including the dTomato gene followed by the SV40 polyA sequence, was amplified by PCR using specific primer oligonucleotides and the p.dTomato plasmid as template (synthesized by Macrogen, Seoul, Republic of Korea). The PCR amplification product was first cloned into the pGEMT-Easy T vector (Promega, Madison, WI, USA) and then subcloned into the BamHI and HindIII sites of the polylinker from the entry vector pENTR4 (Invitrogen, Thermo Fisher Scientific). Then, an in vitro homologous recombination reaction, between the att1 and att2 sites of the entry vector pENTR4.dTomato and the destination adenoviral vector pAdV-CMV/V5-DEST (Invitrogen, Thermo Fisher Scientific), allowed us to obtain the plasmid pAdV-CMV-dTomato. This construct contained the heterologous sequence downstream of CMV promoter and replaced the coding region for the E1 and E3 proteins in the adenovirus genome. The identity of the pAdV-CMV-dTomato was confirmed by nucleotide sequencing (Macrogen). To obtain the recombinant adenoviral vector (AdV.dT), the plasmid pAdV-CMV-dTomato was digested with the restriction enzyme PacI (which allows exposing the ITRs regions) and transfected into HEK293A cultures. This cell line constitutively expressed the product of the E1 viral gene allowing the formation of the infective viral particles. Viral stocks were harvested after the appearance of cytopathic effect and amplified by passages in fresh monolayers. The insertion of the foreign sequence in the recombinant AdV was confirmed by PCR with specific oligonucleotides using total DNA extracted from infected cells as template. Similarly, the expression of the reporter gene was evidenced by microscopic observation of the red fluorescence emitted in infected cells and exposed to ultraviolet light.

In addition, another recombinant AdV that expressed the green fluorescent protein (AdV-GFP) was previously developed and used in this work [27].

The AdVs were purified by sucrose cushion ultracentrifugation as described above. A 5μL aliquot was separated for titration and the rest were frozen at −80 °C. Titration was performed by limiting dilution in HEK293A cells. Viral concentrate was resuspended in PBS and aliquots were quickly frozen at −80 °C. All viral preparations were free from contamination with replicative adenoviruses (RCA) and bacterial lipopolysaccharide (LPS).

### 2.3. Cell Cultures

GBM cell lines C6, U251-MG, and GL26 were grown in Petri dishes containing DMEM with high glucose, L-glutamine, sodium pyruvate, and sodium bicarbonate, supplemented with 10% FBS and 1% penicillin-streptomycin. Cells were harvested using trypsin-EDTA (0.05%) and were counted with Trypan-Blue.

Mouse mIDH Astrocytoma neurospheres were grown in DMEM-F12 media supplemented with 1% Penicillin-Streptomycin, 1x B-27, 1x N-2, 100 ug/mL Normocin, 20 ng/mL bFGF, and 20 ng/mL EGF. Neurospheres were harvested and disaggregated using Accutase and were counted with Trypan-Blue. Both glioma cell lines and murine mIDH neurospheres were kindly provided by Dr. Maria G. Castro from the University of Michigan (MI, US).

The patient-derived gliomas stem cells (GSCs) used in this study were previously isolated from human biopsies following relevant guidelines and national regulations. Cell lines, named G01, G02, G03, G08 and G09, have been described previously [28]. The use of these cultures for biomedical research was approved by the Research Ethics Committee “Comité de Ética en Investigaciones Biomédicas de la Fundación para la Lucha contra Enfermedades Neurológicas de la Infancia (FLENI)”. Patient-derived mIDH and GBM neurospheres were grown in Petri dishes previously coated with Geltrex containing serum-free medium consisting of neurobasal medium supplemented with glucose, sodium pyruvate, PBS-BSA 7.5 mg/mL, 1x B27, 1x N2, 20 ng/mL bFGF and EGF, 2 mM L-glutamine, 2 mM non-essential amino acids, and 50 U/mL penicillin/streptomycin. Cells were harvested using Accutase and were counted with Trypan-Blue.

Primary cultures of mouse astrocytes obtained from the brain cortex were grown in Petri dishes previously coated with gelatin containing DMEM with high glucose, L-glutamine, and sodium pyruvate supplemented with 10% FBS and 1% penicillin-streptomycin. Cells were harvested using trypsin-EDTA (0.05%) and were counted with Trypan-Blue.

Primary cultures of rat astrocytes obtained from the brain cortex were grown in Petri dishes previously coated with borate buffer and polylysine containing DMEM-F12 supplemented with 10% FBS and 1% penicillin-streptomycin. Cells were harvested using trypsin-EDTA (0.05%) and were counted with Trypan-Blue.

### 2.4. Animals

Adult male C57Bl/6 mice (6–8 weeks old) were obtained from the animal facility of the Faculty of Veterinary Sciences, National University of La Plata, Argentina. Mice were maintained under controlled conditions of light (12 h light–dark cycles) and temperature (20–25 °C). They were fed standard feed and water ad libitum and their environment was cared for to minimize stress. All animal experimentation was performed under the guidance of the NIH, and was approved by the Institutional Committee for the Care and Use of Laboratory Animals (CICUAL), School of Medicine, University of Buenos Aires; Res. (CD) No. 697/19 and 2071/15.

### 2.5. Flow Cytometry

Cell lines and neurospheres were harvested 48 h after transduction with AdVs or BVs with 0.025% trypsin-EDTA or Accutase and washed with cold PBS. Cells were then fixed with 4% PFA and resuspended in PBS. Finally, the cells were analyzed by flow cytometry (FACS) in a FACScalibur equipment (Becton Dickinson, Franklin Lakes, New Jersey). Untransduced cells were used to determine the cut-off point for citrine or GFP fluorescence. The analysis of the data obtained by flow cytometry was performed using the FlowJo software v10 program.

### 2.6. Cell Viability

Cell viability was evaluated using 3-(4,5-dimethylthiazol-2-yl)-2,5-diphenyltetrazolium bromide (MTT; Molecular Probes, Invitrogen, Thermo Fisher Scientific) 48 h after transduction with AdV or BV. Absorbance was determined using a 96-well plate spectrophotometer (Bio-Rad, Hercules, CA, Model 550) at 595 nm.

### 2.7. In Vivo Experiments

In order to characterize the transduction efficiency of the vectors in vivo, mice were injected by stereotactic surgery in the brain with 3 doses of BVs (1 × 10^8^ PFU/1 uL; 2.5 × 10^8^ PFU/2 uL, and 5 × 10^8^ PFU/5 uL). We used 2 mice per dose and assessed transgene expression 5 days after injection. We selected the highest dose for the following experiments, since, with the lower doses, we did not observe expression of the reporter protein. Once the dose was chosen, another group of C57Bl/6 mice were injected with AdVs (10^7^ PFU in 3 μL) and BVs (5 × 10^8^ PFU in 5 μL) unilaterally into the right striatum with stereotactic surgery, using a 5 μL Hamilton syringe with a 33-gauge needle (3 animals per vector). The coordinates for the injection of the AdV were +0.5 mm AP; −2.1mm ML; −2.9; −3.2; −3.5 mm (1 uL/point) DV from the bregma. The coordinates for the BV injection were +0.5 mm AP; −2.1 mm ML; −2.9; −3.2; −3.5; −3.8; −4.1 mm DV (1 uL/point) from the bregma. After 5 days, mice received an overdose of ketamine/xylazine and were perfused with heparinized Tyrode’s solution and 4% paraformaldehyde (PFA). Brains were removed and fixed with 4% PFA for an additional 48 h, resuspended in cold 20% sucrose for 16 h, and finally frozen in an acetone-isopentane bath at −80 °C. Cryostat sections of 50 μm were obtained and transduction efficiency was subsequently analyzed by observing the expression of the reporter gene dTomato.

To assess the ability of BV to transduce glioma cells in vivo, we intracranially inoculated mouse mIDH neurospheres into the right striatum of naïve mice (n = 3). Three weeks later, we injected BV intratumorally (5 × 10^8^ PFU in 5 μL) and 5 days later, mice were perfused with heparinized Tyrode’s solution and 4% PFA. Brains were processed as described above to assess transduction efficiency by observing the expression of the reporter gene citrine.

To study the stability of BVs, brains of C57Bl/6 naïve or i.p. BV-preimmunized mice (10^8^ PFU) were inoculated into the right striatum with BV (5 × 10^8^ PFU). Control mice received an i.p. injection of PBS (n = 3 per condition and time point). After 7 or 21 days, brains were processed as mentioned above and the expression of citrine was evaluated by fluorescent microscopy.

Quantification of transduced cells in each brain sample (5 representative sections per mouse) was achieved using ImageJ software v1.52p.

### 2.8. Nissl Staining

Brain sections were mounted on slides treated with vectabond, dehydrated, and stained with 0.25% crystal violet. Then, they were dehydrated and mounted with Canada balsam and photographed using an Olympus DP73 (ZEISS, Jena, Germany, Axio Scope.A1 microscope).

### 2.9. Hematoxylin and Eosin Staining

Brain sections were mounted on slides treated with vectabond, hydrated with distilled water, and stained first with haematoxylin and then with eosin. Then, they were dehydrated, cleared, and mounted with Canada balsam and photographed using an Olympus DP73 (ZEISS Axio Scope.A1 microscope).

### 2.10. Neutralization Plaque Reduction Assay

High Five^TM^ cells were seeded in 48-well plates at 5 × 10^4^ cells per well the day prior to infection. Sera from pre-immunized mice, non-immunized mice, positive control (ɑ-GP64), and negative control (PBS) were serially diluted in a volume of 100 μL and pre-incubated with 50μL of viruses at 28 °C for 1h. Cells were infected with serially diluted serum/virion mixtures and 1 vol. of 1.6% methylcellulose was then added. Resulting fluorescence plaques were counted 72 to 96 h post infection using epifluorescence Nikon Eclipse e200 microscope. Results were expressed as percentage of negative control.

### 2.11. Immunofluorescence

For anti-GFAP and anti-CD45 immunohistochemistry, brain sections were permeabilized with citrate buffer (pH 6) and with TBS-0.5% Triton-0.1% sodium azide. Blocking was performed in TBS-0.2% Triton-0.1% Sodium azide-10% goat serum for 1 h. Then, the tissues were incubated with rabbit anti-GFAP (840001, 1/2000, Biolegend, San Diego, CA, USA) or rat anti-CD45 (103102, 1/200, Biolegend) antibodies overnight in TBS-0.2% Triton-0.1% sodium azide-1% goat serum. The next day, the tissues were incubated with their respective anti-rabbit (Vector Laboratories Inc., Newark, CA, USA) or anti-rat (Biolegend) secondary antibodies and stained with DAPI. Brain sections were mounted on slides using Vectashield (Vector Laboratories Inc., Newark, CA). Negative controls were incubated without the primary antibody. Tissues were analyzed by fluorescent microscopy.

### 2.12. RNA Sequencing and Reverse Transcription Polymerase Chain Reaction

RNA was extracted from patient-derived GBM cells using TRIzol reagent (Thermo Fisher Scientific) and sequenced in an Illumina sequencing platform. A total of 30 million pair-ended reads of 100 pb length were analyzed using a TruSeq Stranded mRNA LT library.

For total RNA extraction, cell cultures were dissociated with TRIzol reagent (Thermo Fisher Scientific) in accordance with manufacturer’s instructions. cDNA synthesis was performed using MMLV reverse transcriptase (Promega, Madison, WI, USA). Quantitative RT-PCR assays were performed using SYBR^®^ Green-ER™ qPCR SuperMix Universal (Thermo Fisher Scientific). Primers used were: HSPG2 forward 5′-CTTCCACAGACTCTTATC-3′, reverse 5′-TAGGATGATGTTGTTACC-3′; SDC1 forward 5′-ATGAAGAAGAAGGACGAAGG-3′, reverse 5′-CAAGGAAGAGGCAAGTGG-3′; CLTC forward 5′-ATTGTCCTTGATAACTCTGTATTC-3′, reverse 5′-TTGCTGATGGCGATATTGG-3′; DNM1 forward 5′-CGTGATGTGCTGGAGAAC-3′, reverse 5′-GTGGCGATAAGATGGATGG-3′. PCR amplifications were performed in a StepOnePlus™ Real-Time PCR System (Applied Biosystems, Foster City, CA, USA).

### 2.13. Statistical Analysis

Data were plotted and analyzed using GraphPad Prism version 8 software (GraphPad Software, Boston, MA, USA). All data were evaluated for normality using the Kolmogorov-Smirnoff test before performing parametric statistical tests. Differences in transduction efficiency were analyzed by analysis of variance (ANOVA) followed by Tukey’s post-test. Correlations were analyzed by the Spearman test. Differences were considered significant when *p* < 0.05. All experiments were performed at least twice.

## 3. Results

### 3.1. Transduction Efficiency of BVs and AdVs in Normal and Neoplastic Astrocytes In Vitro

In order to assess the capacity of BVs to transduce normal and tumoral glial cells, we used an AdV expressing the green fluorescent protein (GFP), which was already developed and tested [27], and constructed an AdV encoding the red fluorescent protein dTomato, as well as two BVs encoding the green fluorescent protein citrine and dTomato, respectively. We used GFP and citrine for green fluorescence indistinctively as both fluorophores are similar in size (~900 vs. ~700 bp), brightness, and excitation and emission spectra [29]. Mice were pre-immunized with the BV expressing dTomato and then challenged intracranially with the BV expressing citrine in order to ensure that the immunity generated was directed against the viral vector and not against the reporter gene.

We incubated the human cell line U251-MG with the BV encoding citrine (BV.citrine) under the control of the CMV promoter. Please note that, for practical purposes, MOI (multiplicity of infection) was used here to indicate infective virus particles (PFU) per cell, regardless of the fact that no infectious cycle proceeds in mammalian cells after exposure to transducing BV. As a control, we used the AdV vector encoding GFP under the control of the CMV promoter at the same MOIs. As previously reported [30], AdV vectors exhibited a very high transduction efficiency in this cell line, easily transducing over 90% of cells at an MOI of 100 PFU/cell and reaching a plateau of almost 100% transduction efficiency at MOI 500 (Figure 1A). Although the transduction efficiency of BV was lower than the AdV, it transduced over 40% of cells at MOI 500, reaching a plateau of 50% of transduced cells at MOI 1000 (Figure 1B).

Although the classification of diffuse gliomas has traditionally been based on their histopathological characteristics, these tumors are currently classified by distinctive genetic and epigenetic alterations [31]. Currently, the mutational status of the enzyme isocitrate dehydrogenase (IDH) 1 and 2 has become the main marker of prognosis and molecular classification in adult diffuse gliomas [32]. While tumors with mutated IDH (mIDH), i.e., oligodendrogliomas and astrocytomas, present better prognosis, wild type IDH (wtIDH) gliomas are all classified as GBM, regardless of their histopathology [31]. Considering that traditional cell lines do not recapitulate the genetic and phenotypical characteristics of human tumors, we next evaluated the ability of BVs to transduce neurospheres derived from biopsies from four GBM (wtIDH) patients (G02, G03, G08, G09) (Figure 2A). Contrary to what we found in the GBM cell line, in these cultures, BV exhibited a very similar transduction efficiency to AdV, which ranged from 12% to 35% positive cells. The process of cell entry of BV into mammalian cells involves the interaction of its glycoprotein GP64 with different molecules expressed on the cell surface, i.e., heparan sulfate proteoglycan 2 (perlecan) (HSPG2), syndecan1 (SDC1), clathrin (CLTC), and dynamin (DNM1) [22,33,34]. Thus, we performed a correlation analysis between the expression of these molecules, as assessed by RNAseq, and the transduction rate of BV in each of the wtIDH neurospheres derived from patients (Figure 2B). Although no correlation was detected for HSPG2, SCD1, and DNM1, the expression of clathrin exhibited a very strong positive correlation with the transduction efficiency of BV in these GBM cultures. The expression of these markers in all patient-derived wtIDH and mIDH glioma cells was also confirmed by qPCR (not shown).

Considering that gene therapy could also benefit the treatment of mIDH glioma patients, we evaluated whether BV transduce neurospheres derived from a mIDH astrocytoma biopsy (G01) (Figure 2C). Both BV and AdV again exhibited similar levels of transduction efficiency in these cells, with ~60% transduction efficiency.

We next aimed to compare the ability of BV to transduce neoplastic vs. normal astrocytes. We assessed the transduction efficiency of these vectors in the rat GBM cell line C6, as well as in primary cultures of astrocytes obtained from the rat brain cortex. BV exhibited very low transduction efficiency in rat GBM cells compared to AdV, with less than 10% vs. ~50% positivity at MOI 1000 (Figure 3A). In contrast, the transduction efficiency of astrocytes with BV was above 50%, even higher than that achieved with AdV at the same MOI (Figure 3B).

With the same approach, we also assessed the transduction efficiency of these vectors in the mouse GBM cell line GL26, as well as in primary cultures of astrocytes obtained from the mouse brain cortex. In GL26 cells, vectors showed a similar behavior than in U251-MG cells. AdV-mediated transduction was very robust, reaching a plateau of 80% positive GBM cells and around 50% positive primary astrocytes at MOI 500 (Figure 4A,B). BV showed a linear dose-dependent increase in transduction efficiency, with 40% positive GBM cells at MOI 1000 and around 40% positive primary astrocytes at MOI 750 (Figure 4A,B). Incubation with both viral vectors, even at the highest viral concentrations, did not generate cytotoxicity per se in murine astrocytes, as the viability of these cells remained above 95% at all MOIs evaluated (Figure 4B). We also assessed the transduction efficiency of BV vs. AdV in neurospheres derived from genetically engineered mIDH astrocytomas developed in mice (NPA-I) [35]. Transduction efficiency followed a linear MOI-dependent increase for both viruses, which reached ~60% positive cells at MOI 1000 (Figure 4C).

### 3.2. Transduction Efficiency of BVs and AdVs in Normal Mouse Brain

Since gene therapy vectors are injected in the non-neoplastic brain of GBM patients upon surgical removal of the main tumor mass, we evaluated the transduction efficiency and neurotoxicity of BVs in comparison with AdVs in naïve mouse brain. The criteria used for the selection of the doses used of each vector was to choose the greatest amount of viral vector without neurotoxicity. We performed a dose escalation for BVs in the mouse brain. We injected mice in the striatum using stereotactic surgery using three doses of the BV encoding citrine (1 × 10^8^ PFU, 2.5 × 10^8^ PFU, and 5 × 10^8^ PFU). We found that the highest dose tested (5 × 10^8^ PFU) transduced brain cells and did not generate neurotoxicity (not shown). Therefore, we used this dose for BV injection into the brain. Considering that it has been previously shown that the maximum tolerated dose for AdVs in the mouse brain is 10^7^ PFU per site [36], we used this dose for intracranial AdV injection.

We injected mice in the brain with the BV (5 × 10^8^ PFU in 5 μL) encoding dTomato under the control of the CMV promoter (Figure 5A) and as positive controls, a group of mice was injected with AdV (10^7^ PFU in 3 μL) encoding the same reporter protein. Transgene expression was readily detected for both vectors 5 days after administration (Figure 5B). Infiltration of immune cells, as assessed by immunofluorescence using an anti-CD45 antibody, was similar in the brains injected with AdV and BV (Figure 5C).

### 3.3. Transduction Efficiency of BVs in Normal and Neoplastic Astrocytes In Vivo

We next aimed to evaluate the ability of BVs to transduce normal and neoplastic astrocytes in vivo. Mice bearing intracranial gliomas were injected intratumorally with BV.citrine (5 × 10^8^ PFU). Transgene expression was detected in tumor cells within the tumor mass and in non-neoplastic cells surrounding the tumor 5 days after BV injection (Figure 6A). In the brain of naïve mice injected with BV.citrine, transgene expression was detected in astrocytes as assessed by immunofluorescence using an antibody against GFAP (Figure 6B).

In order to determine the stability of BV-mediated transgene expression in vivo, we quantified citrine-positive cells in the mouse brain 7 and 21 days after BV-injection. We found that BV transgene expression was stable for at least 21 days in the mouse brain (Figure 6C), without apparent signs of neurotoxicity, as assessed by Nissl and hematoxylin-eosin staining (Figure 6D,E).

### 3.4. Transduction Efficiency of BVs in the Brain of Mice Preimmunized against BVs

Although pre-existing immunity against BVs has not been reported in humans, it is important to evaluate what would happen in patients exposed to these vectors in order to better understand the performance of these viruses as gene therapy vectors. In order to generate a pre-existing immune response against BV, mice were injected i.p. with BV.dTomato (10^8^ PFU), which was repeated two weeks later. Two weeks after the boost, mice were injected in the striatum with BV.citrine (5 × 10^8^ PFU), and transgene expression was evaluated in coronal brain sections after 7 and 21 days. We found that transgene expression was drastically reduced at 7 days after BV i.c. injection in mice that were previously immunized, an effect that was replicated at day 21 (Figure 7A–D).

## 4. Discussion

Gene therapy is a promising strategy for the treatment of GBM. AdVs have been extensively studied in neuro-oncology, showing excellent safety profiles following intracranial injection in GBM patients in multiple trials over the past three decades [6,37,38,39,40]. However, despite the tremendous efforts of translational neuro-oncology researchers, the median survival of these patients has remained virtually unchanged for almost 20 years. Thus, it is crucial to improve the therapeutic approaches for this disease.

Although AdV vectors exhibit excellent transduction efficiency and toxicity profile in the CNS, virtually the entire population has pre-existing immunity against AdVs [38], which leads to the rapid elimination of the vector by the immune system, leading to transient gene expression even in an immunologically privileged tissue such as the CNS [13]. On the other hand, BVs are vectors for which there is no pre-existing immunity in humans, as they are natural pathogens of insects. Thus, BV transduction could be more persistent compared to AdV, especially in the brain. These vectors include additional advantages, such as high cloning capacity and relative ease of production [17,18,19,21,22,41,42]. Their broad tropism and lack of replication in mammalian cells make them an attractive strategy for gene therapy [18,22,25,26,41,43].

BVs have been previously shown to transduce brain cells, both in vitro and in vivo [23,24]. When we performed a dose escalation for BVs in the mouse brain, we found that the highest dose tested (5 × 10^8^ PFU) did not generate neurotoxicity. This is in accordance with a previous article in which the authors injected a dose of 3 × 10^8^ PFU of BV in the brain of naïve mice with good transduction efficiency and an excellent safety profile [23]. BV transduction has been previously shown to be limited to the cuboid epithelium of the choroid plexus in ventricles upon injection in the corpus callosum of the rat brain [23]. However, we found that BV-mediated transduction was present in astrocytes and well distributed within the striatum, a pattern that was very similar to that of AdVs. Our findings are in agreement with Sarkis et. al., who reported that, 7 days after injection in the striatum, BVs transduce mainly glial cells in vivo in the brain of mice [24]. These discrepancies may be related to the species tested or the injection site. Nevertheless, the findings from Lehtolainen et al. also indicate that BV-mediated expression levels and safety were comparable to the AdV-mediated gene delivery [23]. In this study, we observed good BV-mediated transduction efficiency not only in normal astrocytes in vitro and in vivo, but also in human and murine glioma cell lines and neurospheres in vitro and in vivo. Although the dose of BVs required to achieve transduction efficiency in the brain was higher than that of AdVs (5 × 10^8^ PFU), it did not generate apparent neurotoxicity, which is normally the case with AdVs, limiting the amount of virus that can be injected [36]. In fact, it was previously reported that BV-mediated transduction in the rat brain leads to a much lower microglial response when compared to the AdV-injected brain [23]. Considering that we used the maximum tolerated dose for each vector, we did not observe qualitative differences in the immune cell infiltration in the injection site between BV- and AdV-injected brains. Our findings suggest that BVs could be useful tools to deliver therapeutic transgenes into brain cells, with potential prospects for application in degenerative and neoplastic brain disorders.

All therapies that have been developed have failed when used as monotherapy in GBM, even those that work in other tumors, and the survival of GBM patients has not markedly improved since 2005, i.e., when temozolomide was introduced to the standard of care in addition to surgery and radiotherapy [44]. The complex nature of these tumors requires the rational combination of multiple therapeutic approaches. The in-depth characterization of the performance of each vector could be exploited for the design of combined therapies. A first approximation was previously reported by Granio et al., in which a bi-viral complex composed of BV associated with AdV via CAR receptors superficially displayed on the surface of the BV was used to improve transduction efficiency in cells refractory to AdV transduction [45]. BVs have also been proposed to be used as tools to improve the limited tumor-cell infection and intratumoral distribution of oncolytic vectors [46]. BVs expressing the cellular receptor for reoviruses on its envelope were combined with reoviruses. The biviral complexes resulted in improved tropism and increased cytotoxicity of the oncolytic vector in GBM neurospheres that were otherwise reovirus-resistant [46]. Thus, the ability of BVs to distribute within these tumors could improve the efficacy of oncolytic viruses. In fact, in this study, we found that BVs exhibited a remarkable ability to distribute within GBM neurospheres. In addition, GBM treatment could be improved by the BV-mediated delivery of genes that facilitate chemosensitivity, i.e., expression of proapoptotic molecules or silencing of DNA repair enzymes and multidrug resistance proteins, or genes that boost antitumor immunity, i.e., local silencing of multiple immunological checkpoints without the systemic toxicity that could be developed when using systemic inhibiting antibodies. Although BVs exhibited a lower transduction efficiency compared to AdVs in commercial cell lines, the transduction rate was very similar in more representative glioma models, i.e., murine neurospheres derived from genetically induced murine tumors and patient-derived mIDH and wtIDH glioma cells. These results indicate that BVs may be used as an alternative vector platform in neuro-oncology. These findings also highlight the importance of working with models that better recapitulate the genetic and molecular context of gliomas. These systems also have stem cell characteristics and allow for three-dimensional modeling [28].

BVs and AdVs exhibited very similar transduction efficiency in patient-derived glioma cells, but these rates were very variable amongst these neurosphere cultures. It is possible that the ability of BVs to transduce different tumor cells is dependent on the expression of receptors and cell surface molecules that interact with the baculovirus glycoprotein GP64, mediating BV cell entry. In this study, we found that heparan sulfate proteoglycan 2 (perlecan) (HSPG2), syndecan-1 (SCD1), clathrin (CLTC), and dynamin (DNM1) were present in all patient-derived glioma cells. Interestingly, we found a very robust correlation between the expression of clathrin, a protein involved in BV endocytosis, and the transduction efficiency of BVs in these patient-derived GBM cultures. Thus, clathrin could be an interesting marker to predict the ability of BVs to transduce different tumors and tissues. On the other hand, since BVs have a large cloning capacity, they can be genetically modified to improve their transduction efficiency [47]. One strategy involves increasing the expression of the glycoprotein GP64 or through pseudotyping strategies using proteins from other viruses, such as the vesicular stomatitis virus G protein (VSV-G), the neuraminidase protein or hemagglutinin from the influenza virus, which mediate efficient cell entry [41,48,49]. These genetic modifications that take advantage of the versatility and high cloning capacity of BVs could be combined with strategies to direct the vectors to target cells.

A growing body of evidence shows broad tropism and transduction flexibility of BV [41]. Considering that BVs efficiently transduced murine astrocytes in vitro and in vivo without signs of cyto- or neuro-toxicity, these vectors could be used for therapeutic transgene delivery in neurodegenerative disorders. However, this also highlights the fact that transgene expression will not be tumor-specific when injected in the tumor bed after surgical resection of the main tumor mass. This lack of tumor selectivity is not desirable for gene therapy vectors and it is a limitation of all non-replicating vectors used in neuro-oncology. However, the specificity of the strategy could be improved by the correct selection of the therapeutic transgene, i.e., conditionally cytotoxic enzymes, and immune-stimulant transgenes that trigger antitumor immunity. BV-mediated expression of the Diphteria toxin was achieved using the glial cell-specific promoter GFAP, leading to a robust antitumor effect in an experimental rat GBM model [47]. A strategy to limit BV-mediated transgene expression to tumor cells is the use of glioma-specific promoters such as SSX4 and FOS [50,51]. Together, these strategies would increase the transduction efficiency of BVs in tumor cells vs. astrocytes.

Using an immunocompetent murine model of orthotopic glioma, we found that BVs can transduce tumor glial cells in vivo. These results suggest that BV could be used as a gene therapy vector for the expression of therapeutic transgenes in the field of neuro-oncology. However, since BV genome remains episomal, it is expected that its expression in the tumor decreases over time due to the proliferation of tumor cells, as it happens with other vectors that do not integrate, i.e., AdVs. Thus, it is important to use strategies that do not require long-term expression of the vector, i.e., strategies that block immunological checkpoints or tolerogenic molecules such as IDO, IL10, and TGF-β in order to facilitate antitumor immunity. Our findings on the lack of neurotoxicity after intracranial injection of BVs suggest that, if the transgene is not toxic in normal cells, these vectors can be injected into the tumor bed after the surgical removal of the tumor mass without the risk of neurotoxicity.

BV-mediated transgene expression in naïve brain was stable at least for 21 days post-injection. Similar results have been found in other studies where the expression of a reporter transgene in naïve rat brains was apparent for at least 90 days after the recombinant BV injection [52]. Although no pre-existing immunity has been reported, if these vectors reach clinical use, for instance as vaccine platforms, it would be important to establish the stability of BV expression in the brain of mice that were preimmunized with BVs. We demonstrated that transgene expression was almost lost 7 days after the intracranial injection in preimmunized mice. Since we used BVs encoding different reporter transgenes in the preimmunization and the intracranial injection, the loss of expression was due to the recognition of the BV proteins by the immune system rather than the recognition of the reporter transgene, which could also be immunogenic because it is a foreign protein.

The central aim of this work was to explore the performance of BVs not only as tools to treat brain tumors, but also as vectors for gene delivery into the CNS. Although we did not observe higher transduction efficiency for BVs compared to AdVs, we believe that a variety of different vectors for rational therapy design is desirable and beneficial. In addition, due to the well-known capacity of BVs to incorporate large foreign genetic information, other BV-enhancing genes may be incorporated in addition to the therapeutic gene. In this way, BVs with increased transduction efficiency [53,54,55,56,57] and with the ability to display specific tumor receptor ligands on the surface of the virion have been developed [20,58]. These technologies could be exploited to improve the transgene expression and specificity of BVs for the treatment of brain disorders. In addition, there are approaches using BV-based systems that, after transducing mammalian cells, produce replicative episomes, greatly improving transduction and gene delivery, even without active viral replication [55,59,60].

In the last few years, several research groups have been working to overcome the regulatory requirements necessary for the development of clinical phases, generating standard guidelines for large-scale amplification, concentration, purification, and formulation of BVs [41,61,62,63,64]. For example, the Baculogenes project, funded by the European Union, was one of the consortia that developed large-scale production methods suitable for translating these vectors into the clinic [41]. Our work provides empirical evidence indicating that BVs are potentially useful gene therapy tools to transfer genes into the brain, being a cost-effective and safe strategy for the population.

## Figures and Tables

**Figure 1 viruses-15-00608-f001:**
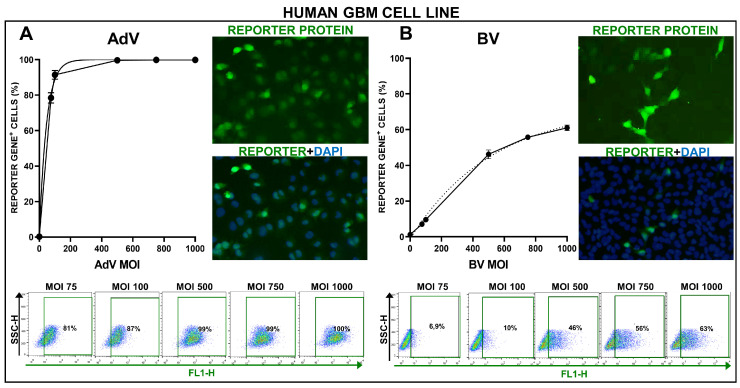
Transduction efficiency of AdV and BV in U251-MG human GBM cell line. U251-MG human GBM cells were incubated with an AdV (**A**) or a BV (**B**) at different MOIs, and 48 h later, transgene expression was assessed by fluorescent microscopy and quantified by flow cytometry. Representative microphotographs are shown (MOI = 750). Nuclei were stained with DAPI. Graphs show the transduction efficiency of each vector at different MOIs, as assessed by flow cytometry.

**Figure 2 viruses-15-00608-f002:**
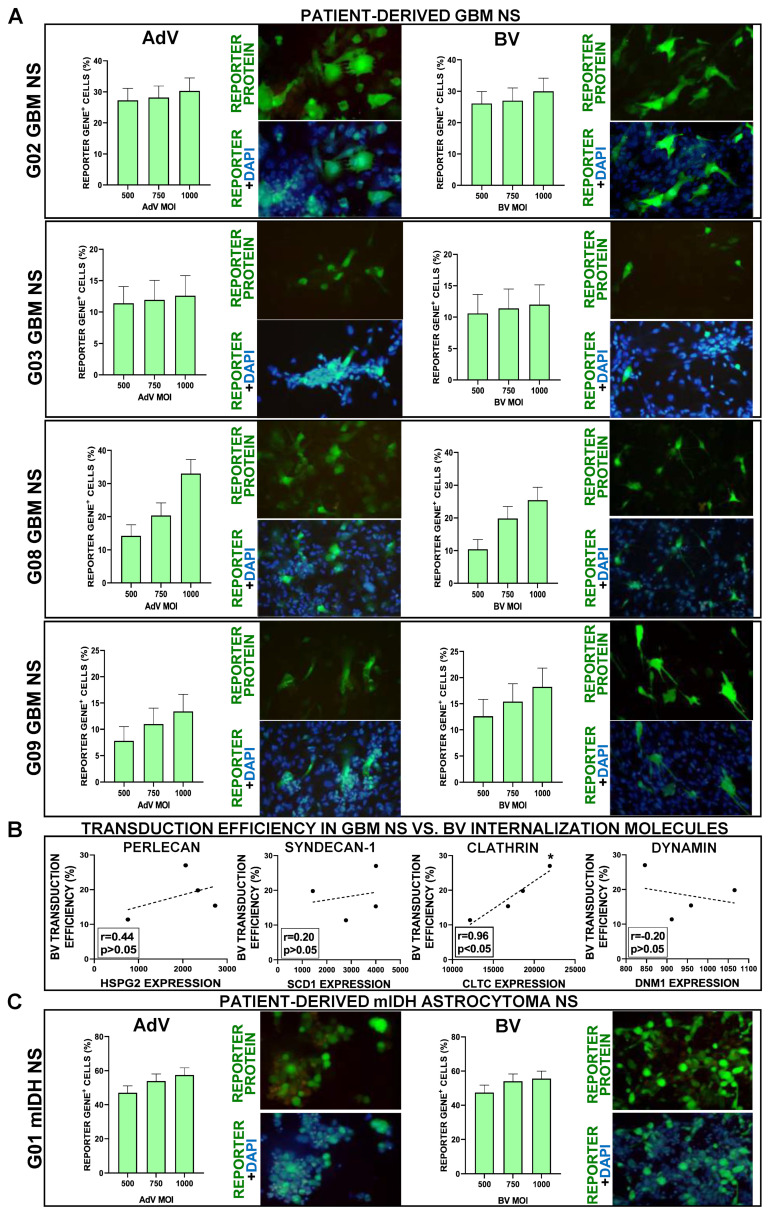
Transduction efficiency of AdV and BV in patient-derived GBM and mIDH astrocytoma neurospheres. (**A**) Patient-derived GBM neurospheres (NS) were incubated with AdV or BV at different MOIs, and 48 h later, transgene expression was assessed by fluorescent microscopy and quantified using software ImageJ. Representative microphotographs are shown (MOI = 750). Nuclei were stained with DAPI. (**B**) Correlation between the transduction efficiency of BVs in patient-derived GBM neurospheres (NS) (MOI = 750) and the expression of molecules involved in BV cell internalization: heparan sulfate proteoglycan 2 (perlecan) (HSPG2), syndecan-1 (SDC1), clathrin (CLTC), dynamin (DNM1). r: correlation coefficient (Spearman correlation test), (**C**) Patient-derived mIDH astrocytoma neurospheres (NS) were incubated with AdV or BV at different MOIs, and 48 h later, transgene expression was assessed by fluorescent microscopy and quantified using ImageJ. Representative microphotographs are shown (MOI = 750). Nuclei were stained with DAPI.

**Figure 3 viruses-15-00608-f003:**
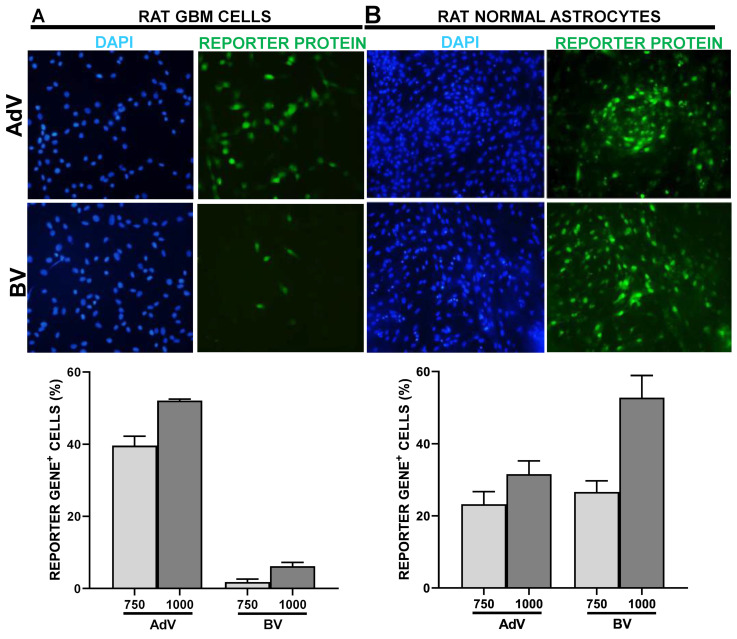
AdV and BV transduction efficiency in rat normal and neoplastic astrocytes in vitro. C6 rat GBM cells (**A**) and primary cultures of normal rat astrocytes from brain cortex (**B**) were incubated with AdV or BV at different MOIs, and 48 h later, transgene expression was assessed by fluorescent microscopy and quantified using ImageJ. Representative microphotographs are shown (MOI = 750). Nuclei were stained with DAPI.

**Figure 4 viruses-15-00608-f004:**
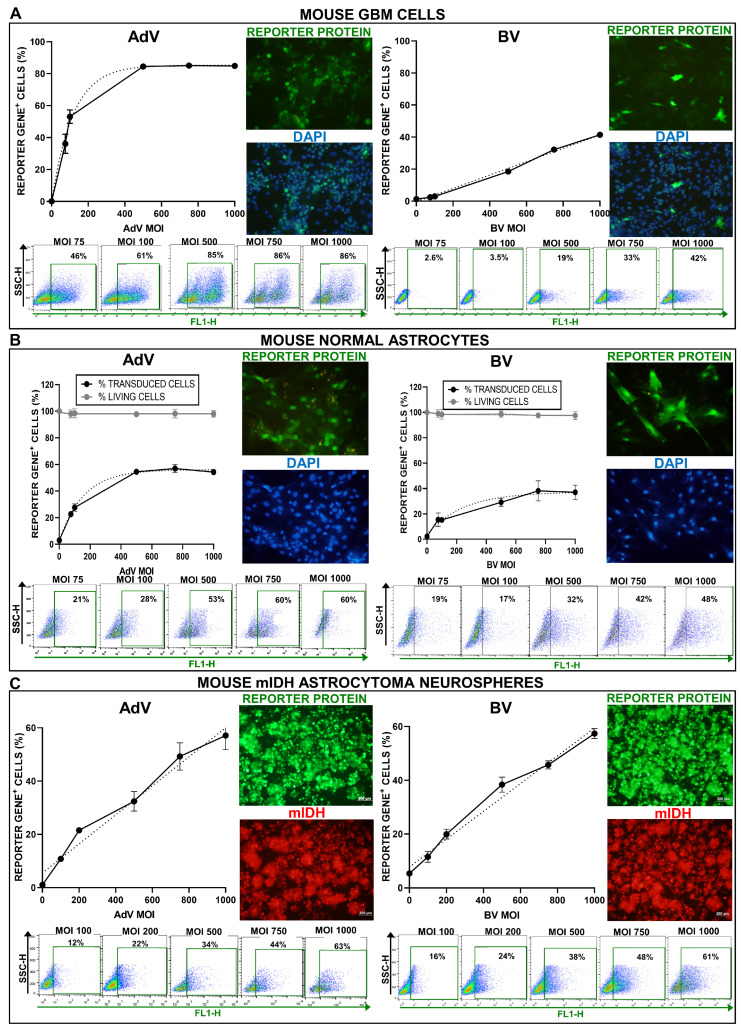
AdV and BV transduction efficiency in mouse normal and neoplastic astrocytes in vitro. GL26 mouse GBM cells (**A**), primary cultures of normal astrocytes from mouse brain cortex (**B**) and Katushka+ mouse mIDH astrocytoma neurospheres (**C**) were incubated with AdV or BV at different MOIs, and 48 h later, transgene expression was assessed by fluorescent microscopy and quantified by flow cytometry. Graphs show the transduction efficiency of each vector at different MOIs, as assessed by flow cytometry. Light gray lines indicate the percentage of living cells at each MOI compared to mock controls, as assessed by MTT assay. Representative microphotographs are shown (MOI = 750). Nuclei were stained with DAPI (**A**,**B**) or Katushka expression (**C**) was used to identify cells.

**Figure 5 viruses-15-00608-f005:**
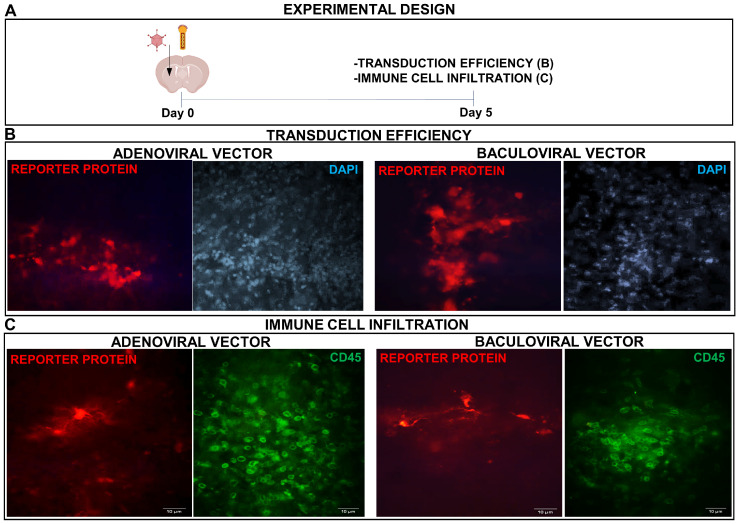
AdV- and BV-mediated transgene delivery in mouse brain. (**A**) C57Bl/6 mice were inoculated into the right striatum with AdV (10^7^ PFU in 3 μL) or BV (5 × 10^8^ PFU in 5 μL). After 5 days, the expression of the reporter protein dTomato (red) was evaluated by fluorescent microscopy and nuclei were stained with DAPI (**B**) or CD45 cells were detected by immunofluorescence (green) (**C**). Representative microphotographs are shown.

**Figure 6 viruses-15-00608-f006:**
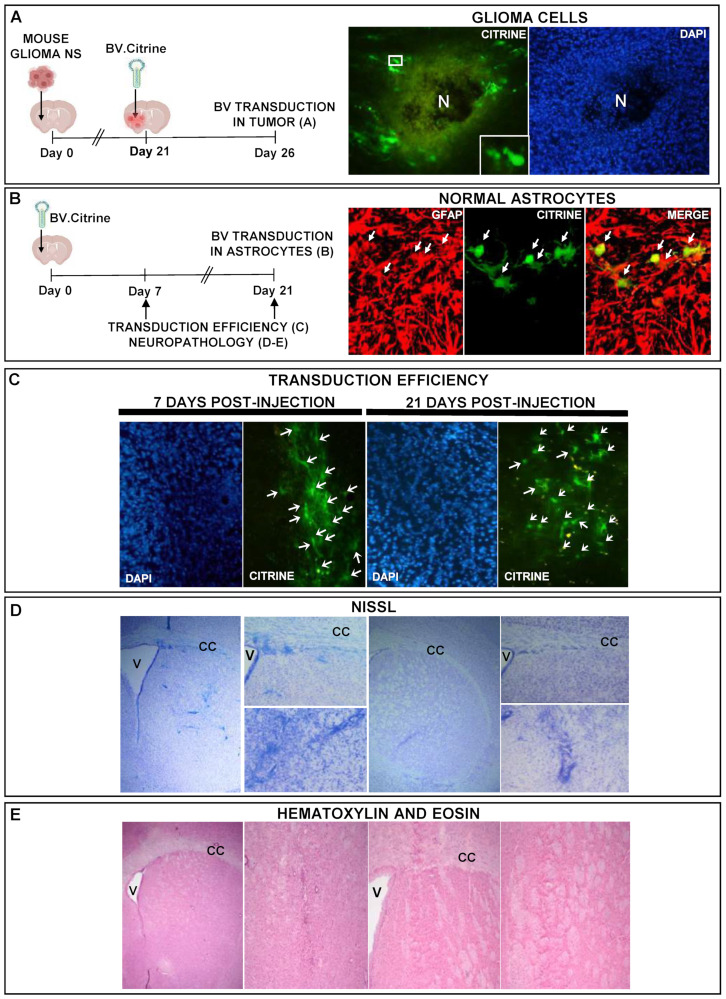
BV-mediated transgene expression in normal and neoplastic astrocytes in vivo. (**A**) C57Bl/6 mice were inoculated with mouse glioma neurospheres (NS) into the brain and 3 weeks later, tumors were injected with BV (5 × 10^8^ PFU) and mice were euthanized after 5 days. The expression of the reporter protein citrine (green) in the tumor was evaluated by fluorescent microscopy. Nuclei were stained with DAPI. The insets show tumor cells with citrine expression, at higher magnification. N indicates necrotic area. (**B**–**E**) C57Bl/6 mice were inoculated into the right striatum with BV (5 × 10^8^ PFU). After 7 or 21 days, the expression of the reporter protein citrine (green) was evaluated by fluorescent microscopy. Glial cells were identified using an anti-GFAP antibody (red stain). The nuclei were stained with DAPI. Arrows indicate glial cells transduced with BV. Neuropathology was assessed by Nissl (**D**) and hematoxylin-eosin staining (**E**). V: Ventricle. CC: Corpus callosum.

**Figure 7 viruses-15-00608-f007:**
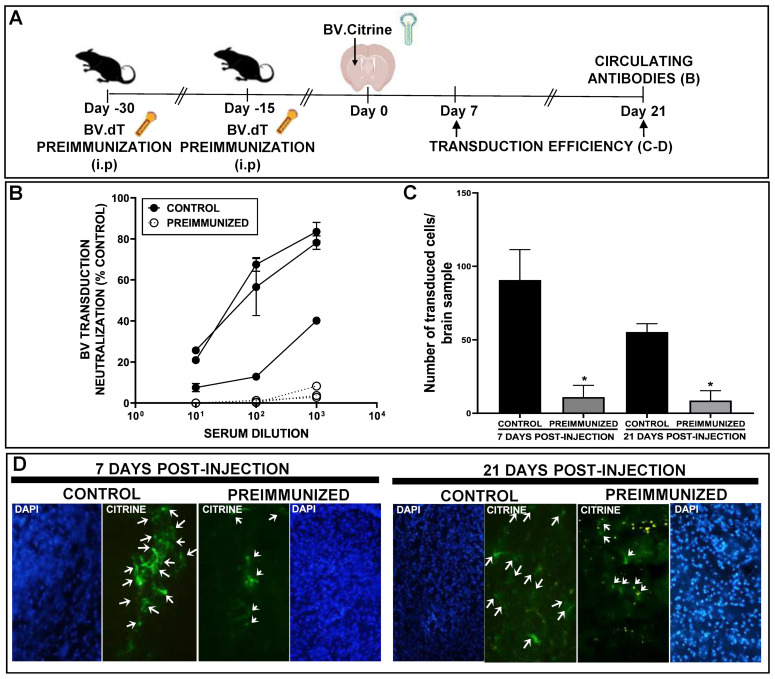
BV-mediated transgene expression stability in the brain of preimmunized mice. (**A**) C57Bl/6 mice were injected i.p with BV.dTomato (10^8^ PFU), 30 and 15 days before administration of BV.citrine (5 × 10^8^ PFU) into the right striatum. (**B**) Sera was extracted from naïve or systemically BV-preimmunized mice and the concentration of anti-BV circulating antibodies was measured by plaque reduction neutralization assay (PRNT). (**C**) 7 or 21 days after intracranial surgery, the expression of citrine (green) was evaluated by fluorescent microscopy and quantified using ImageJ (* *p* < 0.05 vs. control). Representative images of control and preimmunized mouse brains are shown. The nuclei were stained with DAPI. Arrows indicate cells transduced with BV (**D**).

## Data Availability

Data available upon request (gvidela@fleni.org.ar).

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
