# Peer review of "Evaluation of Baculoviruses as Gene Therapy Vectors for Brain Cancer"

_viruses, 2023, doi:10.3390/v15030608_

Round 1

Reviewer 1 Report

Dear authors,

Thank you for submitting your paper. The results you obtained are interesting; however, the extensive revision of your manuscript is needed before the publishing. Please, see my questions and suggestions below.

Majors:

1.     When anti-tumor effect of AdV vectors is described, the importance of AdV ability to replicate, lyse the tumor cells (which results in immunogenic cell death), and spread within the tumor site should be discussed. All the AdV anti-tumor therapeutics developing nowadays are replication competent; thus, the entry and transduction efficacy are just two of many aspects of AdV anti-tumor efficacy. Therefore, you should provide convincing arguments why the anti-glioma therapy by BVs which are not able to replicate and have no superiority in transduction ability compared to AdVs may be considered as an option. Why can the lack of cancer selectivity be beneficial? Why can gene therapy of gliomas be better than virotherapy? What type of gene therapy with BVs besides glioma treatment would work best?

2.     You should clearly describe why you used the different doses and different reporter genes for BVs and AdVs comparison.

3.     Please, provide the figures with better quality and make sure that all the figures are visible (e.g., Figure 5).

Minors:

General

1.     Be consistent with the cell line names U251 (e.g., U251-MG).

2.     In vitro and in vivo should be in italic.

3.     Make sure all abbreviations are spelled out at first mention (CNS, GSC, CMV).

First page:

The title should be written in Sentence case.

All affiliations, KEY WORDS, CORRESPONDING AUTOR – should be written according to the template

Abstract:

Line 32: showed strong correlation with clathrin expression – the brief explanation why you assessed that correlation should be provided.

Introduction:

Line 59: “inclusion-forming units” should be changed to plaque-forming unit if you meant infectious titer and VP/cell if you wanted to provide the physical titer.

Lines 61-64: That’s true for adenovirus type 5; however, alternative AdV types are now using as a delivery vectors which have much lower seroprevalence (e.g., Skog et al, Molecular therapy, 15 (12),  P. 2140-2145).

Materials and Methods:

Please, be consistent while providing the companies’ names and origins.

Line 100 and 102-103: 27,000 rpm and 14,000 RPM – should be provided in g.

Line 108: Cytomegalovirus (CMV) the abbreviation should be first provided earlier (line 88).

Line 136: storage solution -please provide the storage buffer composition

Line 133: why sucrose instead of classical cesium-chloride centrifugation was performed?

Line 149: GSC – spell it out

Line 190 – 195: the experimental design is not clear. How many groups of mice did you use? How many animals per group? Were mice injected with various viral doses of BVs and then with AdVs/BVs? Or was it different groups? Why was the AdV dose lower than BV?

Line 211 and 213: “systemically” and “i.p “ ? Does that mean that experimental and control animals received dose by different routes?

Results:

Please, make sure that all the figures and legends are placed within one page.

Figure 3: it’s better to put the same virus pictures vertically so that the graph for AdV would be placed below the pictures of the AdV.

Figure 5: make sure that the entire figure is visible

Lines 362-366: It’s still unclear the aim of this study. Did you check the influence of pre-existing immunity? If the first experiment was to establish the minimal effective transduction dose, describe how that transduction efficacy was measured. Please, write in clearly.

Lines 365-368: Why were viral doses different?

Lines 369-371: How did you calculate the level of immune cells infiltration to postulate that it was similar?

Figure 6: What stands for CC and V?

Discussion:

Lines 481-482: Using an immunocompetent murine model of orthotopic glioma we found that BVs can transduce tumor glial cells in vivo with high efficiency. What results have led you to call transduction highly efficient? Comparison or quantitative analysis was not carried out.

Author Response

We would like to express our gratitude to the reviewers for their detailed revision of our manuscript. Please, find below the answers to the comments raised. All remarks and comments were incorporated into the new manuscript.

REVIEWER 1

Dear authors,

Thank you for submitting your paper. The results you obtained are interesting; however, the extensive revision of your manuscript is needed before the publishing. Please, see my questions and suggestions below.

Majors:

1.a     When anti-tumor effect of AdV vectors is described, the importance of AdV ability to replicate, lyse the tumor cells (which results in immunogenic cell death), and spread within the tumor site should be discussed. All the AdV anti-tumor therapeutics developing nowadays are replication competent; thus, the entry and transduction efficacy are just two of many aspects of AdV anti-tumor efficacy. Therefore, you should provide convincing arguments why the anti-glioma therapy by BVs which are not able to replicate and have no superiority in transduction ability compared to AdVs may be considered as an option.

We agree with this Reviewer that AdV-mediated oncolytic virotherapy has experienced enormous development in the last decade. In fact, local treatment of GBM patients with oncolytic AdVs has been evaluated in several clinical trials in GBM patients. However, the efficacy of this strategy in GBM patients remains to be determined (Lu, Shah et al. 2021). Recent findings suggest that oncolytic virotherapy may require repeated administration of the vectors in order to exert a clinical benefit. Oncolytic herpes simplex virus G47∆ was intracranially injected up to 6 times over a period of 5 months in 19 GBM patients in a Phase II trial in Japan, leading to a median survival of 20 months (Todo, Ito et al. 2022). Although the improvement in the survival of these patients vs. those treated with standard of care is remarkable, their prognosis remains dismal as the treatment does not cure the disease. Thus, a combination of therapeutic approaches is needed in such a complex disease in order to substantially improve the survival of GBM patients. Non-replicating AdVs have also been recently tested with promising results in GBM patients. The local administration of two AdVs expressing the conditionally cytotoxic enzyme HSV1-TK and the cytokine Flt3L, respectively, has been evaluated in GBM patients in a recent Phase I trial at the University of Michigan (NCT01811992). This strategy, which was well tolerated and stimulated tumor T cell infiltration (Umemura, Orringer et al. 2022), is advancing towards a Phase II trial in a larger cohort of patients. Considering that pre-existing anti-adenoviral immunity by natural infection and/or vaccination impairs long-term AdV-mediated gene expression even in an immune-privileged environment like the brain (Barcia, Jimenez-Dalmaroni et al. 2007; Castro, Candolfi et al. 2014), it is important to increase the availability of gene therapy vectors to treat this disease and other neurological disorders. In fact, an increase in the titers of circulating anti-adenoviral antibodies is observed upon intratumoral injection with therapeutic AdVs in GBM patients (Umemura, Orringer et al. 2022). Thus, our aim was to perform an exhaustive preclinical evaluation of recombinant baculoviruses to deliver therapeutic transgenes into the brain. Since these vectors are natural pathogens of insects, patients do not present pre-existing immunity against them, which could facilitate transgene expression even after repeated intracranial vector injection. Thus, they could be used to deliver genes that facilitate the apoptotic response of tumor cells or that boost antitumor immunity, strategies tested in GBM patients in recent or current clinical trials using non-replicating AdVs (NCT03596086, NCT03603405, NCT02026271, NCT01811992). Larger availability of gene therapy vectors may allow developing therapeutic combinations with improved efficacy to treat these deadly tumors.

It is important to highlight that the central aim of this work was to explore the performance of BVs not only as tools to treat brain tumors, but also as vectors for gene delivery into the CNS. We compared them against AdVs, which are known by their robust transduction efficiency in the CNS and excellent safety profile upon intracranial injection. Although we have not observed higher transduction efficiency for BV compared to AdV, we believe that having a variety of different vectors for rational therapy design is desirable and beneficial. In addition, due to the well-known capacity of BVs to incorporate large foreign genetic information, other BV-enhancing genes may be incorporated in addition to the therapeutic gene. In this way, baculoviruses with increased transduction efficiency (Sung, Chen et al. 2013; Kolangath, Basagoudanavar et al. 2014; Tamura, Kawabata et al. 2016; Wang, Naik et al. 2017; Hu, Li et al. 2019) and with the ability to display specific tumor receptor ligands on the surface of the virion (Makela, Matilainen et al. 2006; Pidre, Arrías et al. 2023) have been developed. These technologies could be exploited to improve transgene expression and specificity of BVs for the treatment of neurological disorders. In addition, there are approaches using BV-based systems that, after transducing mammalian cells, produce replicative episomes, greatly improving transduction and gene delivery even without active viral replication (Lo, Hwang et al. 2009; Sung, Chen et al. 2013; Sung, Chen et al. 2014).

These arguments have now been included in the Introduction (Page 2) and Discussion sections (Pages 17 and 18) in the revised version of our manuscript.

1.b Why can the lack of cancer selectivity be beneficial?

The lack of tumor selectivity is not desirable for gene therapy vectors and it is a limitation of all non-replicating vectors used in neuro-oncology. However, the specificity of the strategy could be improved by the correct selection of the therapeutic transgene, i.e. conditionally cytotoxic enzymes, and immune-stimulant transgenes that trigger antitumor immunity. Nevertheless, the rational design of gene therapy vectors using tumor-specific promoters, or the incorporation of genes into the BV genome that allow tumor targeting, would easily avoid this lack of selectivity and direct the therapy toward tumor cells. In fact, tumor-specific BV-mediated expression of the Diphteria toxin was achieved using a GFAP promoter, leading to a robust antitumor effect in an experimental rat GBM model (Wang, Li et al. 2006).

Of note, although wild-type BVs and other recombinant gene therapy vectors used in neuro-oncology lack tumor selectivity, the ability of these vectors to transduce normal astrocytes without signs of neurotoxicity, as we show here, could be exploited in other brain disorders, such as neurodegenerative diseases.

This has been added to the Discussion section of our revised manuscript (Page 18).

1.c Why can gene therapy of gliomas be better than virotherapy?

In spite of enormous efforts of clinicians, neurosurgeons, and researchers, as well as millions of taxpayer dollars invested in neuro-oncology research, the survival of GBM patients has not been markedly improved since 2005 when Temozolomide was introduced to the standard of care in addition to surgery and radiotherapy (Stupp, Mason et al. 2005). All individual therapies developed have failed, even those that work in other tumors, i.e. immunological checkpoint inhibitors, antitumor vaccines, and targeted therapies. The complex nature of these tumors requires the combination of multiple therapeutic approaches. The in-depth characterization of the performance of each vector could be exploited for the design of combined therapies. A first approximation was previously reported by Granio et. al. in which a bi-viral complex composed of BV associated with AdV via CAR receptors superficially displayed on the surface of the BV was used to improve transduction efficiency in cells refractory to AdV transduction (Granio, Porcherot et al. 2009). BVs have also been proposed to be used as tools to improve the limited tumor-cell infection and intratumoral distribution of oncolytic vectors (Dautzenberg, van den Hengel et al. 2017). BVs expressing the cellular receptor for reoviruses on its envelope were combined with reoviruses and the biviral complexes resulted in improved tropism and increased cytotoxicity of the oncolytic vector in GBM neurospheres that were otherwise reovirus-resistant (Dautzenberg, van den Hengel et al. 2017). Thus, the ability of BVs to distribute within these tumors could improve the efficacy of oncolytic viruses. In addition, BVs could be used for the delivery of transgenes that sensitize tumor cells to chemotherapy and radiotherapy or that reduce tumor immunosuppression to boost the efficacy of immunotherapeutic strategies.

This has now been added to the Discussion section of our revised manuscript (Page 17).

1.d What type of gene therapy with BVs besides glioma treatment would work best?

As depicted above, GBM treatment could be improved by BV-mediated delivery of genes that facilitate chemosensitivity, i.e. expression of proapoptotic molecules or silencing of DNA repair enzymes and multidrug resistance proteins, or genes that boost antitumor immunity, i.e. local silencing of multiple immunological checkpoints without the systemic toxicity that could be developed when using systemic inhibiting antibodies. These strategies could improve the efficacy of the standard of care or immunotherapeutic approaches (antitumor vaccines, CAR-T cell therapy).

  1. You should clearly describe why you used the different doses and different reporter genes for BVs and AdVs comparison.

The criteria used for the selection of the doses was to select the greatest amount of viral vector without neurotoxicity. When we performed a dose escalation for BVs in the brain, we found that the highest dose tested (5x108 PFU) did not generate neurotoxicity. Therefore, we used 5x108 PFU for injection into the brain. This is in accordance with a previous article in which authors injected a dose of 3x108 PFU of BV in the brain of naïve mice with good transduction efficiency and excellent safety profile (Lehtolainen, Tyynela et al. 2002).  Considering that it has been previously shown that the maximum tolerated dose for AdVs in the mouse brain is 107 PFU per site (Gerdes, Castro et al. 2000) we used this dose for intracranial AdV injection. For this reason, the dose of AdV was lower than that used for BV.

Regarding the reporter genes we used an AdV expressing GFP, which was already developed and tested (Romanutti, D'Antuono et al. 2013), and developed an AdV encoding dTomato, as well as two BVs encoding citrine and dTomato, respectively. We used GFP and citrine for green fluorescence indistinctively as both fluorophores are similar in size (~900 vs ~700 bp), brightness, and excitation and emission spectra (Cranfill, Sell et al. 2016). Mice were pre-immunized with the BV expressing dTomato and then challenged intracranially with the BV expressing citrine in order to ensure that the immunity generated was directed against the viral vector and not against the reporter gene.

This has been added to the revised version of our manuscript (Page 16).

  1. Please, provide the figures with better quality and make sure that all the figures are visible (e.g., Figure 5).

We apologize for the low quality of the images in the original submission. We have prepared a zip file with the figures in tiff format (higher quality) to be uploaded with the submission of the revised version of our manuscript. In addition, the figures embedded in the manuscript were also modified.

Minors:

General

  1. Be consistent with the cell line names U251 (e.g., U251-MG).

We have corrected this in the revised version of our manuscript.

  1. In vitroand in vivo should be in italic.

We agree with the reviewer and have now corrected it throughout the manuscript.

  1. Make sure all abbreviations are spelled out at first mention (CNS, GSC, CMV).

Now all the abbreviations are spelled out at first mention.

4.a First page: The title should be written in Sentence case.

This has been corrected.

4.b All affiliations, KEY WORDS, CORRESPONDING AUTHOR – should be written according to the template

Title, affiliations, Key Words, and corresponding author information were corrected according to the template.

4.c Abstract: Line 32: showed strong correlation with clathrin expression – the brief explanation why you assessed that correlation should be provided.

Reviewer’s suggestion was included on page 1 (Abstract).

4.d Introduction: Line 59: “inclusion-forming units” should be changed to plaque-forming unit if you meant infectious titer and VP/cell if you wanted to provide the physical titer.

It was changed to a plaque-forming unit.

4.e Lines 61-64: That’s true for adenovirus type 5; however, alternative AdV types are now using as a delivery vectors which have much lower seroprevalence (e.g., Skog et al, Molecular therapy, 15 (12), P. 2140-2145).

This information and the reference were added to the revised version of our manuscript (Page 2).

4.f Materials and Methods: Please, be consistent while providing the companies’ names and origins.

We apologize for these inconsistencies and have now corrected them.

4.g Line 100 and 102-103: 27,000 rpm and 14,000 RPM – should be provided in g.

We have now corrected this.

4.h Line 108: Cytomegalovirus (CMV) the abbreviation should be first provided earlier (line 88).

This was now corrected.

4.i Line 136: storage solution -please provide the storage buffer composition

AdV stocks were stored in PBS. This was now added to the Materials and Methods section (Page 4).

4.j Line 133: why sucrose instead of classical cesium-chloride centrifugation was performed?

We tested both protocols for viral vector purification: the sucrose cushion and the cesium chloride ultracentrifugation techniques. Although AdVs are traditionally purified using cesium chloride, we aimed to use one procedure for both, AdVs and BVs. We found that upon injection in the brain, sucrose cushion-purified vectors exhibited better neuropathology and was a much less expensive and faster procedure, rendering excellent transduction efficiency and safety in vitro and in vivo for both vectors.

4.k Line 149: GSC – spell it out

We have now spelled GSC.

4.l Line 190 – 195: the experimental design is not clear. How many groups of mice did you use? How many animals per group? Were mice injected with various viral doses of BVs and then with AdVs/BVs? Or was it different groups? Why was the AdV dose lower than BV?

We performed four in vivo experiments as follows: (i) BV dose escalation in naïve mouse brain, which included 2 animals per dose (1x108 PFU/ 1 µl; 2.5x108 PFU/ 2 µl and 5x108 PFU/ 5 µl), transduction efficiency and toxicity were evaluated qualitatively 5 days after vector injection; (ii) the assessment of AdV (107 PFU) vs. BV (5x108 PFU) transduction efficiency in naïve mouse brain included 3 animals per vector, transduction efficiency and immune cell infiltration were evaluated qualitatively 5 days after vector injection; (iii) BV (5x108 PFU) intratumor injection in glioma-bearing mice, transduction efficiency was evaluated qualitatively 5 days after vector injection, and (iv) BV intracranial injection in naive or BV-preimmunized mice at different time points, which included 3 mice per condition and time point, transduction efficiency was evaluated qualitatively 7 and 21 days after vector injection. The criteria used for the selection of the doses was to select the maximum tolerated dose of each virus, as described above. This has now been clarified in the revised version of our manuscript (Page 5).

4.m Line 211 and 213: “systemically” and “i.p“? Does that mean that experimental and control animals received dose by different routes?

We apologize for this mistake. Both groups (experimental and control) received an intraperitoneal injection of either BV.dTomato or PBS, respectively. This has now been corrected in the revised version of our manuscript (Page 5).

4.n Results: Please, make sure that all the figures and legends are placed within one page.

We have now placed the figures and their legends in the same page.

4.o Figure 3: it’s better to put the same virus pictures vertically so that the graph for AdV would be placed below the pictures of the AdV.

We agree with the reviewer and have now rearranged the figure.

4.p Figure 5: make sure that the entire figure is visible

We have now made the entire figure visible.

4.q Lines 362-366: It’s still unclear the aim of this study. Did you check the influence of pre-existing immunity? If the first experiment was to establish the minimal effective transduction dose, describe how that transduction efficacy was measured. Please, write in clearly.

In the BV dose escalation experiment, we injected three different doses (1x108 PFU/ 1 µl; 2.5x108 PFU/ 2 µl and 5x108 PFU/ 5 µl) in the brain of naïve mice (n=2 per dose) and 5 days later we analyzed transduction efficiency using fluorescent microscopy and quantified the positive cells by ImageJ per brain sample. Each brain sample consisted in 5 representative sections of each brain. We selected the highest dose tested as it did not generate neurotoxicity. We compared the performance of the BVs at 5x108 PFU with that of AdVs at 107 PFU, which is the maximum tolerated dose per injection site in the mouse brain (Gerdes, Castro et al. 2000).

This has been clarified in the Materials and Methods (Page 5).

One of the main disadvantages in the use of viral vectors in gene therapy is the transient expression of transgenes due to the rapid elimination of the vector by the immune system. Although the expression of gene therapy in the brain does not trigger primary immune responses due to the immune privilege of this organ, when systemic pre-existing immunity exists, the immune system clears vector-transduced cells even in the brain (Barcia, Jimenez-Dalmaroni et al. 2007; Castro, Candolfi et al. 2014). Although no pre-existing immunity against BVs has been reported in the human population, which is a great advantage for its potential use as a gene therapy vector, it is important to highlight that, for example, vaccine production protocols have already been developed that uses the baculovirus as a vaccine vector, so it seemed important for us to study whether this vector generates an immune response upon systemic administration and how this pre-existing immunity would affect the stability of this vector in the brain. When performing the in vivo assay, we verified the presence of circulating BV-specific neutralization antibodies accompanied by a decrease in the expression of the citrine reporter in the brain of mice preimmunized with BV.dTomato compared to the control mice (Fig. 7B).

4.r Lines 365-368: Why were viral doses different?

The criteria used for the selection of the doses was to select the maximum tolerated dose of each virus, as described above.

4.s Lines 369-371: How did you calculate the level of immune cells infiltration to postulate that it was similar?

In this case we did not quantify the number of infiltrating immune cells, so from a qualitative point of view we infer that the immune cell infiltration at the injection site of both vectors is comparable, but we cannot affirm that one is superior to the other (Page 16).

4.t Figure 6: What stands for CC and V?

CC means corpus callosum and V ventricle. This was now stated in the figure legend.

4.u Discussion: Lines 481-482: Using an immunocompetent murine model of orthotopic glioma we found that BVs can transduce tumor glial cells in vivo with high efficiency. What results have led you to call transduction highly efficient? Comparison or quantitative analysis was not carried out.

We apologize for this mistake. Although it has been previously shown that BVs could transduce GBM cell lines in vitro (Sarkis, Serguera et al. 2000; Wang, Li et al. 2006), this is the first report in which BV transduction is detected in experimental GBM in vivo. Since we did not quantify the number of transduced cells in vivo, we have changed the phrasing as follows “Using an immunocompetent murine model of orthotopic glioma we found that BVs can transduce tumor glial cells in vivo” (Page 18).

REFERENCES

Barcia, C., M. Jimenez-Dalmaroni, et al. (2007). "One-year expression from high-capacity adenoviral vectors in the brains of animals with pre-existing anti-adenoviral immunity: clinical implications." Mol Ther 15(12): 2154-2163.

Castro, M. G., M. Candolfi, et al. (2014). "Adenoviral vector-mediated gene therapy for gliomas: coming of age." Expert Opin Biol Ther 14(9): 1241-1257.

Cranfill, P. J., B. R. Sell, et al. (2016). "Quantitative assessment of fluorescent proteins." Nat Methods 13(7): 557-562.

Dautzenberg, I. J. C., S. K. van den Hengel, et al. (2017). "Baculovirus-assisted Reovirus Infection in Monolayer and Spheroid Cultures of Glioma cells." Sci Rep 7(1): 17654.

Gerdes, C. A., M. G. Castro, et al. (2000). "Strong promoters are the key to highly efficient, noninflammatory and noncytotoxic adenoviral-mediated transgene delivery into the brain in vivo." Mol Ther 2(4): 330-338.

Granio, O., M. Porcherot, et al. (2009). "Improved adenovirus type 5 vector-mediated transduction of resistant cells by piggybacking on coxsackie B-adenovirus receptor-pseudotyped baculovirus." J Virol 83(12): 6048-6066.

Hu, L., Y. Li, et al. (2019). "Improving Baculovirus Transduction of Mammalian Cells by Incorporation of Thogotovirus Glycoproteins." Virol Sin 34(4): 454-466.

Kolangath, S. M., S. H. Basagoudanavar, et al. (2014). "Baculovirus mediated transduction: analysis of vesicular stomatitis virus glycoprotein pseudotyping." Virusdisease 25(4): 441-446.

Lehtolainen, P., K. Tyynela, et al. (2002). "Baculoviruses exhibit restricted cell type specificity in rat brain: a comparison of baculovirus- and adenovirus-mediated intracerebral gene transfer in vivo." Gene Ther 9(24): 1693-1699.

Lo, W. H., S. M. Hwang, et al. (2009). "Development of a hybrid baculoviral vector for sustained transgene expression." Mol Ther 17(4): 658-666.

Lu, V. M., A. H. Shah, et al. (2021). "Clinical trials using oncolytic viral therapy to treat adult glioblastoma: a progress report." Neurosurg Focus 50(2): E3.

Makela, A. R., H. Matilainen, et al. (2006). "Enhanced baculovirus-mediated transduction of human cancer cells by tumor-homing peptides." J Virol 80(13): 6603-6611.

Pidre, M. L., P. N. Arrías, et al. (2023). "The Magic Staff: A Comprehensive Overview of Baculovirus-Based Technologies Applied to Human and Animal Health." Viruses 15(1): 80.

Romanutti, C., A. D'Antuono, et al. (2013). "Evaluation of the immune response elicited by vaccination with viral vectors encoding FMDV capsid proteins and boosted with inactivated virus." Vet Microbiol 165(3-4): 333-340.

Sarkis, C., C. Serguera, et al. (2000). "Efficient transduction of neural cells in vitro and in vivo by a baculovirus-derived vector." Proc Natl Acad Sci U S A 97(26): 14638-14643.

Stupp, R., W. P. Mason, et al. (2005). "Radiotherapy plus concomitant and adjuvant temozolomide for glioblastoma." N Engl J Med 352(10): 987-996.

Sung, L. Y., C. L. Chen, et al. (2013). "Enhanced and prolonged baculovirus-mediated expression by incorporating recombinase system and in cis elements: a comparative study." Nucleic Acids Res 41(14): e139.

Sung, L. Y., C. L. Chen, et al. (2014). "Efficient gene delivery into cell lines and stem cells using baculovirus." Nat Protoc 9(8): 1882-1899.

Tamura, T., C. Kawabata, et al. (2016). "Malaria sporozoite protein expression enhances baculovirus-mediated gene transfer to hepatocytes." J Gene Med 18(4-6): 75-85.

Todo, T., H. Ito, et al. (2022). "Intratumoral oncolytic herpes virus G47∆ for residual or recurrent glioblastoma: a phase 2 trial." Nat Med 28(8): 1630-1639.

Umemura, Y., D. Orringer, et al. (2022). "Combined Cytotoxic and Immune Therapy for Primary Adult High-Grade Glioma." medRxiv: 2022.2011.2004.22281950.

Wang, C. Y., F. Li, et al. (2006). "Recombinant baculovirus containing the diphtheria toxin A gene for malignant glioma therapy." Cancer Res 66(11): 5798-5806.

Wang, C. H., N. G. Naik, et al. (2017). "Global Screening of Antiviral Genes that Suppress Baculovirus Transgene Expression in Mammalian Cells." Mol Ther Methods Clin Dev 6: 194-206.

Reviewer 2 Report

The authors investigated the potential of baculoviral vectors (BV) for brain cancer gene therapy and they found that BVs efficiently transduce glioma cells and astrocytes without apparent neurotoxicity. The experiments were well designed and the writing also very nice. The only question in here is whether this kind of job, BV vectors for brain tumor gene therapy, has been published previously. If yes, please add the references in the introduction; if not, the authors may indicate so.

Author Response

ANSWERS TO REVIEWERS

We would like to express our gratitude to the reviewers for their detailed revision of our manuscript. Please, find below the answers to the comments raised. All remarks and comments were incorporated into the new manuscript.

REVIEWER 2

The authors investigated the potential of baculoviral vectors (BV) for brain cancer gene therapy and they found that BVs efficiently transduce glioma cells and astrocytes without apparent neurotoxicity. The experiments were well designed and the writing also very nice. The only question in here is whether this kind of job, BV vectors for brain tumor gene therapy, has been published previously. If yes, please add the references in the introduction; if not, the authors may indicate so.

A few reports have been published showing that baculoviruses could be possible gene therapy tools in brain cancer (Sarkis, Serguera et al. 2000; Dautzenberg, van den Hengel et al. 2017). BVs have been previously reported to transduce rat and human GBM commercial cell lines (Wang, Li et al. 2006). In addition, tumor-specific BV-mediated expression of the Diphteria toxin was achieved using a GFAP promoter, which is specific for glial cells, leading to a robust antitumor effect in an experimental rat GBM model (Wang, Li et al. 2006). BVs have also been proposed to be used as tools to improve the limited tumour-cell infection and intratumoural distribution of oncolytic vectors (Dautzenberg, van den Hengel et al. 2017). BVs expressing the cellular receptor for reoviruses on its envelope were combined with reoviruses and the biviral complexes resulted in improved tropism and increased cytotoxicity of the oncolytic vector in GBM neurospheres that were otherwise reovirus-resistant (Dautzenberg, van den Hengel et al. 2017). Thus, the ability of BVs to distribute within these tumors could be exploited to improve the efficacy of oncolytic viruses. However, up to now no clinical trial has implemented the use of these viruses as vectors of gene therapy in any pathology in humans. In the last few years, several research groups have been working to overcome the regulatory requirements necessary for the development of clinical phases, generating standard guidelines for large-scale amplification, concentration, purification, and formulation of BVs (Carinhas, Bernal et al. 2009; Felberbaum 2015; Kwang, Zeng et al. 2016; Nasimuzzaman, van der Loo et al. 2018; Schaly, Ghebretatios et al. 2021). For example, the Baculogenes project, funded by the European Union, was one of the consortia that developed large-scale production methods suitable for translating these vectors into the clinic (Schaly, Ghebretatios et al. 2021).

BVs have been previously shown to transduce brain cells, both in vitro and in vivo (Sarkis, Serguera et al. 2000; Lehtolainen, Tyynela et al. 2002).  BV transduction was previously shown to be limited to the cuboid epithelium of the choroid plexus in ventricles upon injection in the corpus callosum of the rat brain (Lehtolainen, Tyynela et al. 2002). However, we found that BV-mediated transduction was present in astrocytes and well distributed within the striatum, a pattern that was very similar to that of AdVs. Our findings are in agreement with Sarkis et. al., who reported that BVs transduce mainly glial cells in the brain of mice 7 days after injection in the striatum (Sarkis, Serguera et al. 2000). These discrepancies may be related to the species tested or the injection site used. Nevertheless, the findings from Lehtolainen et. al. also indicate that BV-mediated expression level and safety were comparable to the AdV-mediated gene delivery (Lehtolainen, Tyynela et al. 2002). Here, we observed good BV-mediated transduction efficiency not only in normal astrocytes in vitro and in vivo, but also in human and murine glioma cell lines and neurospheres in vitro and in vivo. Although the dose of BVs (5x108 PFU) required to achieve transduction efficiency in the brain was higher than that of AdVs, it did not generate apparent neurotoxicity, which is normally the case with AdVs, limiting the amount of virus that can be injected (Gerdes, Castro et al. 2000). In fact, it was previously reported that BV-mediated transduction in the rat brain leads to a much lower microglial response when compared to the AdV-injected brain (Lehtolainen, Tyynela et al. 2002). Our work provides empirical evidence indicating that BV are potentially useful gene therapy tools to transfer genes into the brain, being a cost-effective and safe strategy for the population.

REFERENCES

Carinhas, N., V. Bernal, et al. (2009). "Baculovirus production for gene therapy: the role of cell density, multiplicity of infection and medium exchange." Appl Microbiol Biotechnol 81(6): 1041-1049.

Dautzenberg, I. J. C., S. K. van den Hengel, et al. (2017). "Baculovirus-assisted Reovirus Infection in Monolayer and Spheroid Cultures of Glioma cells." Sci Rep 7(1): 17654.

Felberbaum, R. S. (2015). "The baculovirus expression vector system: A commercial manufacturing platform for viral vaccines and gene therapy vectors." Biotechnol J 10(5): 702-714.

Gerdes, C. A., M. G. Castro, et al. (2000). "Strong promoters are the key to highly efficient, noninflammatory and noncytotoxic adenoviral-mediated transgene delivery into the brain in vivo." Mol Ther 2(4): 330-338.

Kwang, T. W., X. Zeng, et al. (2016). "Manufacturing of AcMNPV baculovirus vectors to enable gene therapy trials." Mol Ther Methods Clin Dev 3: 15050.

Lehtolainen, P., K. Tyynela, et al. (2002). "Baculoviruses exhibit restricted cell type specificity in rat brain: a comparison of baculovirus- and adenovirus-mediated intracerebral gene transfer in vivo." Gene Ther 9(24): 1693-1699.

Nasimuzzaman, M., J. C. M. van der Loo, et al. (2018). "Production and Purification of Baculovirus for Gene Therapy Application." J Vis Exp(134).

Sarkis, C., C. Serguera, et al. (2000). "Efficient transduction of neural cells in vitro and in vivo by a baculovirus-derived vector." Proc Natl Acad Sci U S A 97(26): 14638-14643.

Schaly, S., M. Ghebretatios, et al. (2021). "Baculoviruses in Gene Therapy and Personalized Medicine." Biologics 15: 115-132.

Wang, C. Y., F. Li, et al. (2006). "Recombinant baculovirus containing the diphtheria toxin A gene for malignant glioma therapy." Cancer Res 66(11): 5798-5806.

Reviewer 3 Report

The current manuscript under review authored by Fallit et. al. has been very well designed and written. They have done a good job in establishing the need of alternative treatment to AdV GBM and other brain cancerous conditions. The authors have planned appropriate in-vitro and in-vivo studies to introduce BV as an alternating treatment that can have potentially fewer toxicity issues and can have a better potential to deliver gene therapy due to the absence of pre-existing immunity.

The methods, results and discussion sections have been clearly written and contain appropriate information. The only minor edit I have recommended is to fix the Figure 5 alignment as part of figure is being cut out of margin.

Author Response

ANSWERS TO REVIEWERS

We would like to express our gratitude to the reviewers for their detailed revision of our manuscript. Please, find below the answers to the comments raised. All remarks and comments were incorporated into the new manuscript.

REVIEWER 3

The current manuscript under review authored by Fallit et. al. has been very well designed and written. They have done a good job in establishing the need of alternative treatment to AdV GBM and other brain cancerous conditions. The authors have planned appropriate in-vitro and in-vivo studies to introduce BV as an alternating treatment that can have potentially fewer toxicity issues and can have a better potential to deliver gene therapy due to the absence of pre-existing immunity.

The methods, results and discussion sections have been clearly written and contain appropriate information. The only minor edit I have recommended is to fix the Figure 5 alignment as part of figure is being cut out of margin.

We thank this Reviewer for the thorough revision of our manuscript. We are thrilled to hear that this Reviewer likes our work and finds our manuscript “very well designed and written”. We have now rearranged figure 5, as requested by this Reviewer.

Round 2

Reviewer 1 Report

Dear authors,

I appreciate the great work you have done to improve the manuscript and address all the reviewers’ concerns. The manuscript needs only minor revision to be published. Please, verify that inserted paragraphs in the discussion part do not break the flow and that these paragraphs do not look like the reply to reviewers’ concerns but as a discussion of your data and comparison of your data with the previous ones.

Minors:

490-496 – this explanation looks like the answer to the reviewer concern rather than discussion paragraph. It would be better to move that information into correspondent part of Results as it clarify the choice of fluorophores, thus being a technical detail.

The same is true about lines 483-489 when you describe the doses. Some of that information should be rephrased and put into result part. Only results and comparison with the previous data should be left for the discussion.

Lines 498-500: BVs have been previously shown to transduce brain cells, both in vitro and in vivo [23, 24]. BV transduction was previously shown to be limited to the cuboid epithelium of  the choroid plexus in ventricles upon injection in the corpus callosum of the rat brain  -please, be consistent with the verb tenses (have been previously shown/ was previously shown).

Line 520 – what does “individual” mean in this case? All individual therapies developed so far

Author Response

REVIEWER 1

Dear authors, I appreciate the great work you have done to improve the manuscript and address all the reviewers’ concerns. The manuscript needs only minor revision to be published. Please, verify that inserted paragraphs in the discussion part do not break the flow and that these paragraphs do not look like the reply to reviewers’ concerns but as a discussion of your data and comparison of your data with the previous ones.

We agree with this Reviewer and have now modified the location of the inserted paragraphs (highlighted), as indicated below.

Minors:

  1. 490-496 – this explanation looks like the answer to the reviewer concern rather than discussion paragraph. It would be better to move that information into correspondent part of Results as it clarify the choice of fluorophores, thus being a technical detail.

We have now moved this paragraph to the Results section (Line 303) and adapted the text and the list of references accordingly.

  1. The same is true about lines 483-489 when you describe the doses. Some of that information should be rephrased and put into result part. Only results and comparison with the previous data should be left for the discussion.

We have now moved this paragraph to the Results section (Line 405) and adapted the text and the list of references accordingly.

  1. Lines 498-500: BVs have been previously shown to transduce brain cells, both in vitro and in vivo [23, 24]. BV transduction was previously shown to be limited to the cuboid epithelium of the choroid plexus in ventricles upon injection in the corpus callosum of the rat brain -please, be consistent with the verb tenses (have been previously shown/ was previously shown).

We apologize for this oversight and have now corrected it (Line 519).

  1. Line 520 – what does “individual” mean in this case? All individual therapies developed so far.

It refers to therapies that have been used as monotherapies. This has now been clarified in the text (Line 540).
